

# Potential oxygen consumption and community composition of sediment bacteria in a seasonally hypoxic enclosed bay

Fumiaki Mori[1,2,3], Yu Umezawa[4], Ryuji Kondo[5], Gregory N. Nishihara[1,2] and Minoru Wada[1]

[1] Graduate School of Fisheries and Environmental Sciences, Nagasaki University, Nagasaki, Nagasaki, Japan
[2] Institute for East China Sea Research, Organization for Marine Science and Technology, Nagasaki University, Nagasaki, Nagasaki, Japan
[3] Kochi Institute for Core Sample Research, Japan Agency for Marine-Earth Science and Technology (JAMSTEC), Kochi, Japan
[4] Department of Environmental Science on Biosphere, Tokyo University of Agriculture and Technology, Tokyo, Japan
[5] Department of Marine Science and Technology, Fukui Prefectural University, Fukui, Japan

Corresponding author
Fumiaki Mori, morifu@jamstec.go.jp

## ABSTRACT

The dynamics of potential oxygen consumption at the sediment surface in a seasonally hypoxic bay were monitored monthly by applying a tetrazolium dye (2-(4-iodophenyl)-3-(4-nitrophenyl)-5-phenyl-2H-tetrazolium chloride [INT]) reduction assay to intact sediment core samples for two consecutive years (2012–2013). Based on the empirically determined correlation between INT reduction (INT-formazan formation) and actual oxygen consumption of sediment samples, we inferred the relative contribution of biological and non-biological (chemical) processes to the potential whole oxygen consumption in the collected sediment samples. It was demonstrated that both potentials consistently increased and reached a maximum during summer hypoxia in each year. For samples collected in 2012, amplicon sequence variants (ASVs) of the bacterial 16S rRNA genes derived from the sediment surface revealed a sharp increase in the relative abundance of sulfate reducing bacteria toward hypoxia. In addition, a notable shift in other bacterial compositions was observed before and after the INT assay incubation. It was *Arcobacter* (Arcobacteraceae, Campylobacteria), a putative sulfur-oxidizing bacterial genus, that increased markedly during the assay period in the summer samples. These findings have implications not only for members of Delta- and Gammaproteobacteria that are consistently responsible for the consumption of dissolved oxygen (DO) year-round in the sediment, but also for those that might grow rapidly in response to episodic DO supply on the sediment surface during midst of seasonal hypoxia.

## INTRODUCTION

Surface sediment plays an important role in the oxygen cycle of benthic ecosystems due to the large number of microbes (bacteria) and oxygen reactive reductants it contains. A study by *Rivera et al. (2010)* estimated that up to 81% of the total oxygen consumption below the pycnocline could be ascribed to sediment oxygen consumption (SOC) in coastal seas. Measuring SOC *in situ* has been widely conducted to quantify the extent of sediment metabolism and has contributed to expanding our understanding of aquatic bottom environments (*Zimmerman & Benner, 1994*; *Rowe, Kaegi & Morse, 2002*). However, the direct measurement of SOC may underrepresent the biogeochemical processes that contribute to oxygen dynamics in a hypoxic region, as it typically declines under hypoxic conditions due to the lower availability of oxygen in the surface sediment. When dissolved oxygen (DO) in the water overlying sediment becomes anoxic, oxygen consumption is not practically measurable. However, anaerobic heterotrophic bacteria mediate the degradation of organic matter, and the resultant metabolic products, including ammonium ($NH_4^+$), iron (II) ($Fe^{2+}$), and sulfide ($HS^-$), continue to accumulate under the anoxic condition (*Canfield, Thamdrup & Hansen, 1993*; *Middelburg & Levin, 2009*). Thus, a buildup of the reduced compounds leads to increased SOC whenever the bottom DO levels recover. It is, therefore, important to quantitatively evaluate the impacts of both aerobic and anaerobic respiratory processes in the sediment over a wide range of DO conditions from normoxia to anoxia.

SOC can be divided into two components: biological oxygen consumption (BOC) and chemical oxygen consumption (COC). The former reflects aerobic respiration by sediment microbes, while the latter reflects the chemical reactions between reduced compounds and oxygen. Although the relative importance of BOC within SOC has long been suggested (*Jørgensen, 1977*), methods used to determine the individual contributions of sediment BOC and COC have yet to be fully developed. Several previous oceanographic and limnological studies have attempted to explicitly determine COC and BOC by comparing SOC rates of fresh and fixed sediment core samples (*Seiki et al., 1994*; *Zimmerman & Benner, 1994*). However, it often resulted in overestimation of sediment COC due to pH change or the increased penetration depth of oxygen induced by fixatives (*Sweerts et al., 1991*; *Zimmerman & Benner, 1994*).

In our previous studies, the tetrazolium dye reduction assay for non-destructed sediment samples was applied to estimate SOC in a seasonally hypoxic inner bay (Omura Bay, Japan). We assumed two electrons were used to reduce one mole of 2-(4-iodophenyl)-3-(4-nitrophenyl)-5-phenyl-2H-tetrazolium chloride (INT), which is equivalent to the reduction of 1/2 $O_2$ (*Wada et al., 2012*). We further demonstrated that the potential COC (pCOC) from formalin-fixed sediment samples comprised on average 72% of the potential whole oxygen consumption (pWOC) (*Mori et al., 2015*). In some cases, a greater pCOC than pWOC was found, suggesting an effect of fixatives. Another intriguing finding was the apparent increase in potential BOC (pBOC) during severe hypoxia. As the pBOC was estimated by subtracting pCOC from pWOC, the actual pBOC may be greater than that estimated, due to the enhancement of pCOC via formalin fixation. Given the fact that

 

aerobic bacterial activity is suppressed under oxygen depletion *in situ*, it is likely facultative anaerobes are responsible for the increased pBOC. These results raised questions regarding the activity and composition of the microbial (bacterial) community responsible for the SOC in the bay.

However, there are some constraints to be addressed in the methodology used before exploring ecologically relevant questions any further, as the SOC results inferred from the assay were far below those obtained with other direct methods. One possible reason is the development of an oxygen boundary layer over the sediment surface during the incubation assay (*Inoue & Nakamura, 2009*). Another possible reason is the potential toxicity of INT (*Martínez-García et al., 2009*). Although a good positive correlation between INT reduction and oxygen consumption of aerobically incubated sediment samples has been reported (*Trevors, 1984*), extensive comparisons between the data obtained with this method and those from direct measurement of SOC must be conducted to quantitatively address the microbial respiratory processes in the sediment.

Linking SOC dynamics with microbial community composition is key to understanding the biogeochemical processes in sediment ecosystems. It was recently suggested that members of the chemolithoautotrophic sulfur oxidizing bacteria make a substantial contribution to SOC in seasonally hypoxic sediment environments (*Seitaj et al., 2017*). A facultative chemolithoautotrophic Woeseiaceae, a predominant and core member of microbial communities in sediments worldwide, possesses cytochrome c oxidases for aerobic respiration and is therefore likely to make a significant contribution to SOC dynamics (*Mußmann et al., 2017*). Furthermore, members of chemolithotrophic cable bacteria have also been shown to make a large contribution to the total annual oxygen uptake in seasonally sulfidic sediment (*Seitaj et al., 2017*). In our previous studies of Omura Bay, we found a shift in sediment bacterial community structure that occurred as DO levels in the bottom water changed (*Mori et al., 2018a*; *Mori et al., 2018b*). Members of the aerobic Gammaproteobacterial group decreased as oxygen became depleted, while members of the sulfate-reducing Deltaproteobacterial group increased and became predominant. The predominance of sulfate-reducing bacteria enhances the accumulation of hydrogen sulfide ($H_2S$) in sediment, which migrates upward to reach the overlying water and consumes oxygen through abiotic and/or biotic processes. However, further details of the bacterial contribution to SOC dynamics are yet to be determined.

The primary objective of this study was to reveal the dynamics of potential oxygen consumption at the sediment surface of Omura Bay and to uncover the bacterial community composition that contributed to the pBOC. First, we compared the extent of INT reduction with that of oxygen consumption determined using direct measurement of DO in sediment slurry. Based on the correlation between them, relative contribution of pBOC and pCOC to pWOC (SOC) was inferred for a central site of the bay during two consecutive years (2012–2013). We further assessed changes in the bacterial 16S rRNA gene sequences during incubation for the INT reduction assay.

## MATERIALS AND METHODS

### Study site description and sampling procedure

Omura Bay is in Nagasaki Prefecture, western Kyushu, Japan. It comprises steep cliffs at the margins and a large, flat seafloor at a water depth of 15–20 m (*Kato et al., 2003*). The bay is extremely enclosed, is connected to the open sea (East China Sea) by just two narrow channels. Cold and dense oxygenated water flows into the bottom of the bay in winter through these channels (*Nogami et al., 2000*; *Takahashi et al., 2009*). In early summer, warmer and less dense water intrudes into the middle layer of the bay, leaving the bottom water below the depth of intrusion stagnant, consequently leading to hypoxia in the central region of the bay (*Nogami et al., 2000*; *Takahashi et al., 2009*). In autumn, usually by October, this hypoxic water mass disappears as the vertical convection becomes activated by cooling of the sea surface (*Suzaki, Miyake & Nakata, 2013*).

Oceanographic monitoring and sediment core sampling were conducted at a central site in Omura Bay (Station 21; 20 m water depth; 32°55.390′N, 129°51.350′E; see *Mori et al., 2018b*). During the summer months of 2012 and 2013 (Table 1), sediment samples were hand-collected with an acrylic pipe (length 31 cm, inner diameter 26-mm) whilst scuba diving. Surface areas with visible bioturbation were not included in the samples. Temperature, salinity, DO, and chlorophyll-a fluorescence in the water column at the sampling site were measured using a multi-parameter monitoring device (AAQ, JFE-Advantec Co, Kobe, Japan). All of the sediment core samples were stored at an *in situ* temperature and carefully transported to the laboratory within 3 h of collection, avoiding direct exposure to sunlight and other physical disturbances. For the DNA analysis, the top layers (0–5 mm) of three sediment cores were pooled and stored at −20 °C until DNA extraction was performed.

For a series of experiment to compare INT reduction and oxygen consumption (slurry experiments, see below), additional cores were collected using a gravity core sampler (Ashura, Rigosha Co. Ltd, Japan), equipped with three polycarbonate tubes (length 57 cm, inner diameter 82 mm), in August 2017 and 2020. Each sediment core was sliced at a depth 0–5 cm, the sliced sediments were pooled on board the training ship, *Kakuyo-Maru,* and then stored at −80 °C until further processing.

For the time-course experiments, another set of additional sediment cores were collected from the same sampling site of Omura Bay (St. 21) using an acrylic tube (length 31 cm, inner diameter 26 mm), whilst scuba diving on August 7 and September 8, 2014.

### Comparison between the extent of a tetrazolium dye reduction and oxygen consumption of the sediment samples (slurry experiments)

The sediment samples collected in 2017 and 2020 were used to determine the relationship between INT reduction and oxygen consumption in sediment. The tetrazolium dye reduction assay was based on the reduction of INT by the dehydrogenase activity of sediment microorganisms and reduced compounds such as sulfide in the sediment (*Wada et al., 2012*). To reveal the quantitative relationship between INT reduction and oxygen consumption by the sediment samples, we performed a series of incubation experiments using sediment slurries. A freshly thawed slurry sample was prepared by suspending

**Table 1 Dissolved oxygen (DO) conditions in the bottom water and potential oxygen consumption in the sediment of Omura Bay, Japan.** Environmental parameters and some INT reduction rate data were obtained from *Mori et al. (2015)*; *Mori et al. (2018b)*. DO concentration was divided into four categories, according to *Wright, Konwar & Hallam (2012)*; Oxic (>90 $\mu M O_2$), dysoxic (90–20 $\mu M O_2$), suboxic (20–1 $\mu M$ $O_2$), and anoxic (<1 $\mu M$ $O_2$).

| Year | Month/Day | Bottom water | | | | Sediment | | | | | | |
|---|---|---|---|---|---|---|---|---|---|---|---|---|
| | | DO conditions | DO ($\mu M$)[a] | Temperature (°C)[a] | Salinity (PSU)[a] | $pWOC_{INT}$ ($\mu mol\, g^{-1}\, day^{-1}$) | $pCOC_{INT}$ ($\mu mol\, g^{-1}\, day^{-1}$) | pBOC ($\mu mol\, g^{-1}\, day^{-1}$) | TOC ($mg\, g^{-1}$)[a] | TON ($mg\, g^{-1}$) | Sulfide ($mg\, S\, g^{-1}$)[a] | Bacterial cells abundance (Cells $g^{-1}$)[a] |
| 2011 | 7/5 | Dysoxic | 70.4 | 20.1 | 32.6 | 0.57[b] | 0.69[b] | −0.12 | 24.1 | 2.6 | NA | $6.3 \times 10^9$ |
| | 7/15 | Dysoxic | 65.2 | 21.0 | 31.2 | 0.86[b] | 0.62[b] | 0.24 | NA | NA | NA | $6.0 \times 10^9$ |
| | 8/12 | Suboxic | 4.8 | 24.5 | 30.3 | 2.05[b] | 1.14[b] | 0.91 | 26.1 | 2.9 | NA | $7.2 \times 10^9$ |
| | 9/14 | Dysoxic | 44.3 | 25.6 | 28.9 | 0.89[b] | 0.86[b] | 0.04 | 25.7 | 3.0 | NA | $7.2 \times 10^9$ |
| | 11/9 | Oxic | 128.7 | 22.0 | 31.3 | 1.08[b] | 0.68[b] | 0.40 | 24.8 | 2.8 | NA | $7.6 \times 10^9$ |
| | 12/21 | Oxic | 243.3 | 14.0 | 31.1 | 0.91[b] | 0.68[b] | 0.23 | 26.7 | 3.3 | NA | $4.3 \times 10^9$ |
| 2012 | 5/19 | Oxic | 157.6 | 15.7 | 32.9 | 1.16 | 0.89 | 0.27 | 31.3 | 3.7 | NA | $6.1 \times 10^{10}$ |
| | 6/20 | Oxic | 124.0 | 19.0 | 33.1 | 0.98 | 1.18 | −0.20 | 32.2 | 3.8 | NA | $3.4 \times 10^{10}$ |
| | 7/18 | Dysoxic | 82.7 | 22.1 | 31.9 | 1.30 | 1.29 | 0.01 | 33.1 | 3.4 | NA | $3.0 \times 10^{10}$ |
| | 8/24 | Anoxic | 0.0 | 24.1 | 31.3 | 6.44 | 3.62 | 2.82 | 30.6 | 3.6 | NA | $1.7 \times 10^{11}$ |
| | 9/20 | Oxic | 195.9 | 27.3 | 31.2 | 4.70 | 4.01 | 0.69 | 22.0 | 2.9 | NA | $1.7 \times 10^{11}$ |
| 2013 | 6/6 | Oxic | 138.2 | 17.6 | 32.9 | 1.64 | 0.54[b] | 1.10 | 37.8 | 4.3 | 0.12 | $2.5 \times 10^{10}$ |
| | 6/28 | Dysoxic | 70.7 | 20.8 | 31.3 | 0.82 | 0.40[b] | 0.42 | 36.5 | 3.9 | 0.06 | $3.0 \times 10^{10}$ |
| | 7/12 | Dysoxic | 53.2 | 22.7 | 32.8 | 2.49 | 1.57[b] | 0.91 | 39.5 | 4.1 | 0.02 | $3.2 \times 10^{10}$ |
| | 7/26 | Suboxic | 1.6 | 22.3 | 30.4 | 2.90 | 2.86[b] | 0.04 | 37.4 | 4.1 | 0.32 | $3.1 \times 10^{10}$ |
| | 8/2 | Suboxic | 1.8 | 22.8 | 32.7 | 3.35 | 3.62[b] | −0.28 | 38.3 | 4.1 | 0.37 | $2.5 \times 10^{10}$ |
| | 8/12 | Suboxic | 8.0 | 24.6 | 32.6 | 3.50 | 3.87[b] | −0.37 | 35.3 | 3.9 | 0.52 | $2.8 \times 10^{10}$ |
| | 8/28 | Suboxic | 3.1 | 26.4 | 32.7 | 11.39 | 7.23[b] | 4.16 | 31.8 | 3.7 | 0.44 | $3.2 \times 10^{10}$ |
| | 9/12 | Suboxic | 2.9 | 26.9 | 28.5 | 5.40 | 6.70[b] | −1.29 | 36.4 | 3.8 | 0.43 | $2.7 \times 10^{10}$ |
| | 10/31 | Oxic | 166.7 | 22.1 | 30.6 | 2.12 | 1.12 | 1.01 | 37.0 | 4.2 | 0.03 | $3.2 \times 10^{10}$ |

**Notes.**

$pWOC_{INT}$, INT reduction rate measured as an index for potential whole oxygen consumption; $pCOC_{INT}$, INT reduction rate measured as an index for potential chemical oxygen consumption; $pBOC_{INT}$, reduction rate measured as an index for potential biological oxygen consumption; TOC, total organic carobn; TON, total organic nitrogen; NA, not applicable.

[a]Previously published environmental parameters from *Mori et al. (2015)* and *Mori et al. (2018b)*.

[b]Previously published INT reduction rate measured as an index for potential oxygen consumption from *Mori et al. (2015)*.

sediment (approximately 3 g) into 1 L of aerated and filter-sterilized (0.22 μm) artificial seawater (Tetra Marine Salt Pro, Tetra Japan, Tokyo, Japan). The sediment slurry was further divided into two, one was incubated at 20 °C for 20 h without aeration prior to the assay, and the other was directly used for the assay without such a pre-incubation. Afterward, the samples were stored at room temperature and aeration was provided for different periods of time, for up to 48 h, to adjust the amount of reduced compounds in the samples. The sediment slurries were aliquoted into three glass bottles (100 ml each) to which INT (final concentration 0.01%) was added. Each of these samples was gently mixed with a stirrer for 24 h at 26 °C under dark conditions. Following this incubation, the slurries were centrifuged at 1,610× g for 10 min, and the supernatants were filtered through a cellulose acetate membrane filter (pore size 0.22 μm, Advantec, A020A025A). Both the filter and the pellet (sediment) were stored at below −20 °C until analysis. The reduced form of INT (INT-formazan [INTF]) was extracted from the filters and sediments according to the method of *Wada et al. (2012)*. The INTF concentration in the extract was calculated by applying a standard curve with five different known concentrations (ranging from 0.5 to 50 μM) of pure INTF (Sigma-Aldrich, USA), dissolved in isopropanol for the filter and methanol for the sediment. We used the following equation to calculate the whole INT reduction rate:

$$\text{INTF } \mu\text{mol g}^{-1}\text{day}^{-1} = \frac{A+B}{D_w} \times \frac{1}{T} \tag{1}$$

where $A$ and $B$ are the amounts of INTF extracted from the filter and the sediment, respectively; $D_w$ is the dry weight of the sediment used for the extraction of INTF; and $T$ is the time of incubation.

Actual oxygen consumption by the sediment slurries was measured directly using an optode sensor (FireSting O2 fiber-optic oxygen meter; Pyro Sciences, Aachen, Germany) in parallel with the whole INT reduction measurement. Oxygen concentrations in the glass bottles were recorded every 30 s. The sediment slurry was incubated in duplicate under the same conditions as described for the assay of the sediment cores (26 °C, under dark conditions, 24 h). Ultra-pure MQ water (10 mL) was added to each of the slurry samples instead of the aqueous INT solution. Oxygen concentrations in the bottles were recorded for 24 h after the initiation of measurement. To estimate the oxygen consumption rate, a generalized additive model (GAM) was fitted to the changes in DO during the incubation period (Eq. 2).

$$y = f(x) + b \tag{2}$$

Here, y is the DO, x is time, f(x) is the smoothing function, and b is the intercept.

Instantaneous consumption rates were determined by taking the first-order derivative of the fitted GAM. These rates were integrated to obtain the total consumption rate during the experiment. The GAM was fitted using the mgcv package (*Wood, 2003*) in R (version 3.5.0), assuming a Gaussian distribution, an identity link function, and a thin plate spline as the smoothing function. The upper limit for the degrees of freedom of f(x) was 10.

For the sediment fixed with formaldehyde, we also measured the rate of chemical oxygen consumption (COC) and the non-biological (chemical) reduction of INT. Five samples of

non-pre-incubated or pre-incubated sediment slurries (20 °C for 20 h without aeration) were prepared for the assay. Afterward, the samples were fixed by adding 10 mL formalin (formaldehyde: 3.7% w/v final concentration). After 15 min, INT (final concentration 0.01% w/v) and ultra-pure MQ water were added to the fixed slurries. They were incubated at 26 °C under dark conditions for 24 h. The COC of the slurry samples was determined in duplicate using the optode, as described above. INTF was extracted from the slurries in triplicate, and chemical INT reduction rate was determined in the same way as described for whole INT reduction. In an additional experiment, sediment slurries (non-pre-incubation sample) were supplied with sterile air for up to 24 h before formalin fixation to adjust the concentration of reduced compounds in the sediment. A generalized linear model (GLMs) assuming a gamma distribution with a log-link function was applied to analyze the entire set of data (Eq. 3).

$$y \sim Gamma(\mu, \sigma)$$
$$\log(\mu) = b_0 + b_1 x \tag{3}$$

The model coefficients are $b_0$ and $b_1$, $\mu$ is the expected value on the log-link scale, $x$ is the INTF, $\sigma$ is the shape parameter, and $y$ are the observations (dissolved oxygen consumption rate).

## Potential sediment oxygen consumption rate in omura bay estimated by an INT reduction assay

Potential oxygen consumption by the sediment samples was estimated by applying a tetrazolium dye reduction assay method to the intact sediment cores, as the overlying water was artificially replenished with oxygen (*Wada et al., 2012*). The whole INT reduction rate was estimated using the method of *Wada et al. (2012)*. Briefly, triplicate cores were used for the measurement of whole INT reduction rate. Triplicate or duplicate cores, collected in 2012 and 2013, were used for the measurement of chemical INT reduction at each sampling time. The overlying water was replaced with 40 mL of aerated and filter-sterilized (0.22 μm) artificial seawater and 5 mL of MQ water and then gently mixed by pipetting. For chemical INT reduction measurement, 5 mL formalin was added instead of MQ water. After 10 min, 5 mL of 0.1% INT solution (w/v) was added to the overlying water and again gently mixed by pipetting. Upon replacing the water, special care was taken to avoid disturbing the sediment surface. Core samples in the acrylic pipes from each sampling time were incubated in the laboratory at 26 °C under dark conditions for 24 h. Following incubation, the overlying water was siphoned into a sterile plastic tube and filtered through a cellulose acetate membrane filter (diameter 25 mm, pore size 0.22 μm, Advantec, A020A025A). The filters were stored below −20 °C until analysis. The sediment cores were vertically extruded and sliced into horizontal sections of 0–5 mm depth. The sediment slices were stored at less than −20 °C. INTF extraction was performed as described by *Wada et al. (2012)*. The INTF concentration in the extract was calculated by applying a standard curve derived from five different concentrations (ranging from 0.5 to 50 μM) of pure INTF (Sigma-Aldrich), dissolved in isopropanol and methanol for the water and sediment samples, respectively. The concentrations of INTF (μmol g$^{-1}$ day$^{-1}$) was calculated by combining the INTF

values in the overlying water and those of the sediment as described in the above section. In order to simplify the calculation of INT reduction rate per unit mass of sediment ($\mu$mol g$^{-1}$ day$^{-1}$), we assumed the upper most sediment of 0–5 mm depth was responsible for the whole INT reduction in the overlying water of the core samples. Besides, the published data of INT reduction in 2011 (*Mori et al., 2015*) was integrated into the following statistical analysis. For the 2012 samples, the solvent extracted sediment residue was pooled and stored in each sampling month (May through September) at $-20\,^{\circ}$C until DNA extraction.

## Time-course experiments

In order to validate the incubation time for the INT-reduction assay of *Wada et al. (2012)*, triplicate core samples were collected in 2014 in the same way as in 2012 and 2013, incubated in the laboratory at 26 $^{\circ}$C under dark conditions and stopped after different incubation times (0, 2, 4, 6, and 24 h). Following incubation, INTF extraction and the calculation were made as described above.

## Principal Component Analysis (PCA) of the environmental parameters and correlation analysis of the pWOC, pBOC, and pCOC activities

Previously published data for the *in situ* environmental parameters (temperature, salinity, DO, and chlorophyll-a fluorescence in the bottom water and the quantities of sediment organic carbon, and nitrogen) (*Mori et al., 2015*; *Mori et al., 2018b*) were compiled for the PCA. The PCA allowed us to reduce the multicollinearity inherent in the environmental variables, and the first three principal components (PCs) were used as covariates for the ensuing GLMs to examine correlations between with pWOC, pCOC, and pBOC. (Eq. 4).

$$y \sim Gamma(\mu, \sigma)$$
$$\log(\mu) = b_0 + b_1 PC1 + b_2 PC2 + b_3 PC3 \tag{4}$$

The model coefficients are $b_0$, $b_1$, $b_2$, and $b_3$, $\mu$ is the expected value on the log-link scale, $\sigma$ is the shape parameter, and $y$ are the observations (pWOC, pBOC, and pCOC).

PCA and the subsequent GLM analyses were performed using the prcomp and glm functions in R (version 3.5.0). In the visualization steps, the "factoextra" (*Kassambara & Mundt, 2020*), "broom" (*Robinson, Hayes & Couch, 2020*), and "ggplot2" (*Wickham, 2016*) packages were used.

## Illumina MiSeq 16S rRNA gene amplicon sequencing

Microbial genomic DNA in the sediment samples was extracted using an ISOIL DNA extraction kit (Nippon Gene, Tokyo, Japan), according to the manufacturer's instructions. A total of ten genomic samples obtained from the surface sediment layer (0–5 mm depth) at the five sampling times in 2012, collected before and after incubation for the INT reduction measurement, were amplified for Illumina MiSeq 16S rRNA gene amplicon sequencing. Primers specific for the V3–V4 regions of the 16S rRNA gene, 341F and 805R (*Herlemann et al., 2011*), were used. The PCR was conducted in a 25-$\mu$L reaction mixture using a PCR buffer, 5 $\mu$l forward primer (1 $\mu$M), 5 $\mu$l reverse primer (1 $\mu$M), 12.5 $\mu$l 2X Kapa HiFi HotStart ReadyMix (Kapa Biosystems, Boston, MA, USA), and 10

ng of the extracted DNA. Then, 27 cycles of PCR with an initial step of 95 °C for 3 min were run, followed by 95 °C for 30 s, 55 °C for 30 s, 72 °C for 30 s, and finally, 72 °C for 5 min. The size of the PCR products was verified by agarose gel electrophoresis, and the PCR products were further purified with Agencourt AMPure XP (Beckman Coulter, Indianapolis, USA), according to the manufacturer's instructions. A second PCR reaction was performed on the purified PCR products (2.5 µl) to index each of the samples. Two indexing primers (Illumina Nextera XT 72 indexing primers, Illumina, California, USA) were used per sample. The second PCR was carried out in a 25-µL reaction mixture using a PCR buffer, 5 µl forward index primer (1 µM), 5 µl reverse index primer (1 µM), 12.5 µl 2X Kapa HiFi HotStart ReadyMix (Kapa Biosystems), and 2.5 µl of the first PCR products. The second PCR products were then also purified (Agencourt AMPure XP; Beckman Coulter). Following purification, all samples were sent to the Bioengineering Lab. Co., Ltd. for massively parallel 300-bp paired-end sequencing, which was performed on an Illumina MiSeq analyzer. Processing and quality control of the reads was performed using R. For raw sequence data, the adapter sequences were trimmed using the "quasR" package (*Gaidatzis et al., 2015*). Following quality control processes, inclusive of chimera removal, the removeBimeraDenovo, filterAndTrim and dada function in the "DADA2" package version 1.6 (*Callahan et al., 2016*) were used, with default settings. Finally, the amplicon sequence variants (ASVs) (*Callahan, McMurdie & Holmes, 2017*) were classified from kingdom to genus level using the SILVA 132 SSU Ref NR 99 database (*Quast et al., 2013*), resulting in the construction of an ASV table with read counts for the ASVs in all samples. In the visualization step, the "phyloseq" (*McMurdie & Holmes, 2013*) and "ggplot2" packages were used to create stacked bar and plots. To delineate a hierarchical clustering plot, the ASV table was standardized based on coverage-based rarefaction (*Chao & Jost, 2012*). Thereafter, the distance matrix was calculated based on the Bray–Curtis dissimilarity, and clusters were calculated using Ward's method. The "vegan" package (*Oksanen et al., 2019*) and hclust function were used to calculate the dissimilarity and clusters.

All associated raw data can be found at the DDBJ Sequence Read Archive under accession numbers DRA008748 (DRX177081–DRX177090).

# RESULTS AND DISCUSSION

## Comparison between the extent of tetrazolium dye reduction and oxygen consumption of the sediment samples (slurry experiments)

The whole INT reduction rates in the sediment slurries ranged from 7.4 to 28.5 µmol INTF $g^{-1}$ $day^{-1}$, while actual oxygen consumption rates ranged from 44.3 to 390.5 µmol $O_2$ $g^{-1}$ $day^{-1}$ ($n = 16$). The chemical INT reduction ranged from 0.6 to 25.6 µmol INTF $g^{-1}$ $day^{-1}$, while oxygen consumption from the same samples ranged from 38.8 to 147.1 µmol $O_2$ $g^{-1}$ $day^{-1}$ ($n = 25$) (Fig. 1, Table S1). We found a significant relationship between INT reduction rate and oxygen consumption rate in Omura Bay sediment (Eq. 3 $P < 0.0001$, Fig. 1). If we estimate the values of the pWOC based on the GLM model, it ranged from 45.9 to 91.1 µmol O $g^{-1}$ $day^{-1}$, which can be converted to areal unit using density of

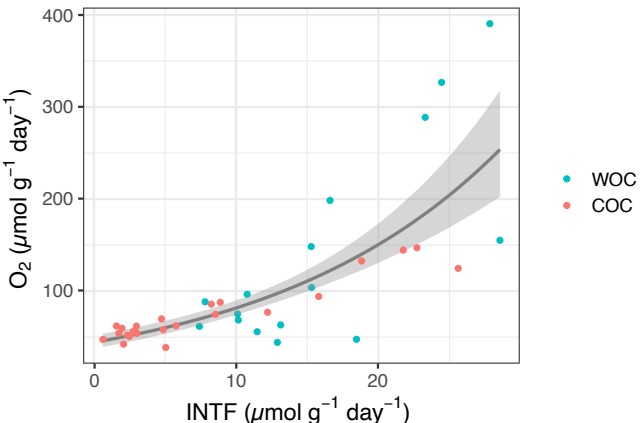

**Figure 1** **INTF formation rate ($\mu$mol g$^{-1}$ day$^{-1}$) vs. oxygen consumption rate ($\mu$mol g$^{-1}$ day$^{-1}$) for whole sediment oxygen consumption (WOC) and chemical oxygen consumption (COC).** The generalized linear model (GLM) used a gamma distribution with a log-link function (Eq. (3), $P < 0.0001$). The fitted lines represent the mean response, and the shaded area indicates the 95% quantiles of the marginal distribution.

sediment and the area of sediment core (45.3 to 56.1 mmol $O_2$ m$^{-2}$ day$^{-1}$). These pWOC values were comparable with the WOCs reported for the basins of other coastal areas, such as the East China Sea (3.6–17.6 mmol $O_2$ m$^{-2}$ day$^{-1}$, *Song et al. (2016)*), the northern Gulf of Mexico (0.82–56.4 mmol $O_2$ m$^{-2}$ day$^{-1}$, *Rowe, Kaegi & Morse (2002)*), and the Seto Inland Sea (3.6–17.6 mmol $O_2$ m$^{-2}$ day$^{-1}$, *Nakamura (2003)*). Therefore, the INT reduction method should work reasonably well to estimate SOC in coastal environments.

The extent of INT reduction has been proven to be an reliable proxy to estimate the oxygen consumption rate of planktonic microorganisms (*Martínez-García et al., 2009*; *Villegas-Mendoza, Cajal-Medrano & Maske, 2019*; *García-Martín et al., 2019*), despite the inherent toxicity of INT (*Martínez-García et al., 2009*; *Villegas-Mendoza, Cajal-Medrano & Maske, 2015*). Previous studies also showed a positive correlation between INT reduction and oxygen consumption for estuary sediments (*Trevors, 1984*) and terrestrial soils (*Trevors, Mayfield & Inniss, 1982*). However, none of them have addressed whole INT reduction and chemical INT reduction separately or determined their correlation coefficients with SOC. Thus, to the best of our knowledge, the present results are the first to demonstrate the relationship between INT reduction and oxygen consumption in marine sediment with a statistically significant GLM model that would allow us to estimate potential SOC based on the INT reduction rates.

## Time-course experiments

We found that the quantity of INTF increased consistently for up to 24 h without appreciable decline in selected sediment core samples (Fig. S1). This was in stark contrast to what had been reported to the planktonic samples. *Martínez-García et al. (2009)* and others (*Villegas-Mendoza, Cajal-Medrano & Maske, 2015*; *García-Martín et al., 2019*) demonstrated that prolonged incubation times can cause an underestimation of the INT reduction rate due to breakage of microbial cells that accumulated INTF crystals inside their cellular membrane.

The seemingly discrepancy between sediment and planktonic samples may be attributable to better extraction efficiency of INTF from sediment samples. Regardless of cell breakage, majority of INTF crystals formed in sediment microbial cells were likely to be retained on the sediment surface and were directly extracted from the sediment slices with methanol. In contrast, INTF crystals formed in planktonic samples could be lost if the size of INTF grain was smaller than that of pore size of the filters. Furthermore, the time for INT in overlying water to reach and exhibit toxicity on microbial cells in the sediment may be delayed due to a slow diffusion of INT across the boundary layer that would develop over the sediment surface under the static condition of the assay.

## Temporal variability of potential SOC

Based on the GLM model obtained with the slurry experiments, the potential SOC in the intact core samples ranged from 45.3 to 56.1 mmol $O_2$ $m^{-2}$ $day^{-1}$. However, as most of the whole INT reduction in the core samples turned out to be far below the minimum value in the slurry experiments (Fig. 1, Table S1), we decided not to describe and discuss the results any further based on the conversion the INT reduction rates into SOC, but to use whole, chemical and biological INT reduction as proxies for potential WOC (pWOC$_{INT}$), COC (pCOC$_{INT}$), and BOC (pBOC$_{INT}$), respectively.

Figure 2 showed the three components of SOC at the central part of Omura Bay during early summer to late autumn in 2012 and 2013 (Figs. 2A–2C, Table 1) together with those in 2011 that had been already published (*Mori et al., 2015*) in order to make the downstream statistical analysis more robust. Both pWOC$_{INT}$ and pCOC$_{INT}$ tended to be greater in August and September than in other sampling months. According to the criteria used by *Wright, Konwar & Hallam (2012)*, we defined hypoxic conditions as <90 $\mu M O_2$, and divided the DO level into four categories: Oxic (>90 $\mu M O_2$), dysoxic (90–20 $\mu M O_2$), suboxic (20–1 $\mu M O_2$), and anoxic (<1 $\mu M O_2$). The potential activity of SOC was highest under suboxic or anoxic conditions in the bottom water. As the highest pBOC$_{INT}$ in 2012 and 2013 (2.82 $\mu mol$ INTF $g^{-1}$ $day^{-1}$ in 2012 and 4.16 $\mu mol$ INTF $g^{-1}$ $day^{-1}$ in 2013) coincided with the timing of the highest bacterial count in each year (1.7 $\times$ $10^{11}$ cells/g sediment in 2012 and 3.2 $\times$ $10^{10}$ cells/g sediment in 2013, Table 1), it was likely that the increases in facultative anaerobes or aerobes contributed to the surge of the pBOC$_{INT}$. In five out of 20 cases, pCOC$_{INT}$ exceeded pWOC$_{INT}$, and pBOC$_{INT}$ showed negative values (Table 1, Figs. 2A–2C). This may be partly explained by the effect of formalin addition on SOC as described above, which lead to an enhancement of COC in the sediment.

In order to identify the environmental driver(s) for the variation of potential SOC, we first conducted PCA for all the environmental variables examined (DO concentration, temperature, and salinity in the bottom water within 0.5 m above the seafloor; and TOC and TON contents in the sediment), during 2011 through 2013 (Table 2, Fig. 3A) and tested the correlation between the obtained principal components (PCs) and the potential SOC (Table 3). It was indicated that the first three PCs explained 93% of the variability of the variables (Table 2). The results of PC1 suggest that TOC as well as TON in the sediment increased in the opposite directions from DO and temperature, reflecting a tendency of organic matter preservation in the sediment under hypoxia. On the other

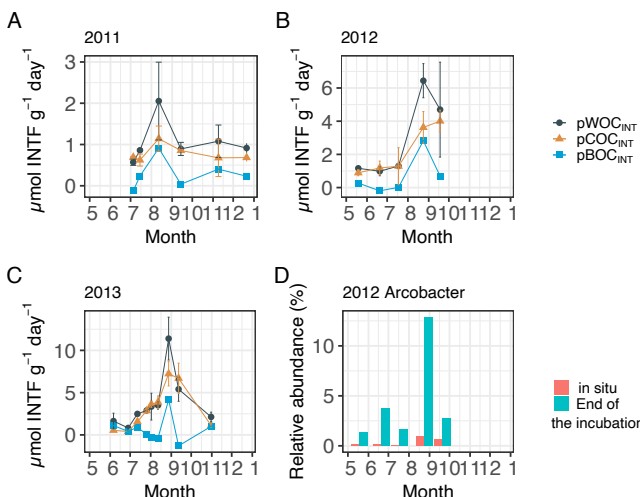

**Figure 2** Sediment oxygen consumption rate (mmol INTF m$^{-2}$ day$^{-1}$) measured with the INT reduction method for (A) 2011, (B) 2012, and (C) 2013. Error bar of potential WOC (pWOC$_{INT}$) represents the standard error ($n = 3$). Error bar for potential COC (pCOC$_{INT}$) represents the range of duplicate measurements from July 2011 through July 2012. Error bars of pCOC$_{INT}$ represent the standard error after August 2012 ($n = 3$). Note that the $y$-axis scale is among figures. Potential sediment oxygen consumption for 2011 and pCOC$_{INT}$ for 2013 were calculated using the INT reduction rate obtained from *Mori et al. (2015)*. (D) shows the relative abundance of the *Arcobacter* group before and after 24 h of re-oxygenation incubation in 2012.

**Table 2** Descriptive statistics of the principal component (PC) scores of environmental paramters.

| PCs | | | | Environmental factors | | | | |
|---|---|---|---|---|---|---|---|---|
| | Standard deviation | Proportion of variance | Cumulative proportion | DO (µM) | Temperature (ºC) | Salinity (PSU) | TOC (mg g$^{-1}$) | TON (mg g$^{-1}$) |
| PC1 | 1.46 | 0.43 | 0.43 | −0.221 | −0.084 | 0.248 | 0.668 | 0.66 |
| PC2 | 1.35 | 0.36 | 0.79 | −0.595 | 0.666 | −0.444 | 0.068 | −0.017 |
| PC3 | 0.85 | 0.14 | 0.93 | 0.430 | −0.181 | −0.829 | 0.183 | 0.248 |

hand, the results of PC2 suggest that temperature increased in the opposite directions from salinity and DO, reflecting a temporal variation of those variables in hypoxic water mass overlying the sampling location. GLM analysis showed that only PC2 had a significant effect on pWOC$_{INT}$ and pCOC$_{INT}$ (Table 3, Figs. 3B, 3D). These results strongly suggest that temperature as well as DO in the bottom water acts as environmental drivers that control the potential SOC.

## Bacterial community composition and its response to re-oxygenation

A total of 2,546 unique ASVs were identified in the surface sediment from the center of Omura Bay in 2012. The dominant taxa in the bacterial communities were affiliated with the Deltaproteobacteria, Gammaproteobacteria, and Bacteroidia (Fig. 4). Within the Deltaproteobacteria class, the family groups of Desulfobacteraceae (major genera Sva0081 sediment group and *Desulfobacter*) and Desulfobulbaceae (major genera *Desulfopila* and

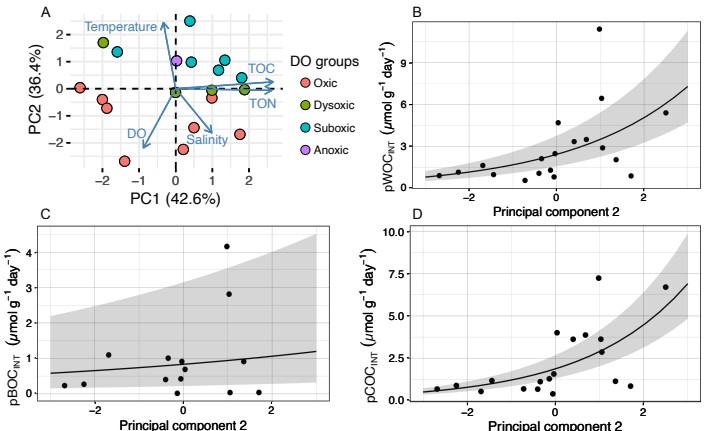

**Figure 3** **The results of principal component analysis (PCA) and a generalized linear model (GLM) using abiotic environmental factors and pSOC.** The GLM used a gamma distribution with a log-link function. The fitted lines represent the mean response, and the shaded area indicates the 95% quantiles of the marginal distribution. (A) The first two principal components. The arrow length indicates the variance of the variable, and the arrow direction indicates increasing values. Arrows pointing in the same direction indicate positive correlations, and arrows pointing in opposite directions indicate negative correlations. The degree of correlation is indicated by the angle between any two arrows, where a right- angle indicates no correlation. (B) $pWOC_{INT}$ and (D) $pCOC_{INT}$ were only significantly affected by the second principal component ($pWOC_{INT}$, $P = 0.007$; $pBOC_{INT}$, $P = 0.570$; $pCOC_{INT}$, $P = 0.001$). Negative $pBOC_{INT}$ values were omitted when GLM analysis was applied.

**Table 3** **Results of generalized linear models for pSOC and PCs.**

| Response variable | Explanatory variable | Estimate | Std.error | *t* value | *p* value |
|---|---|---|---|---|---|
| | PC1 | 0.071 | 0.108 | 0.660 | 0.519 |
| $pWOC_{INT}$ | **PC2** | **0.366** | **0.116** | **3.145** | **0.007** |
| | PC3 | −0.220 | 0.185 | −1.185 | 0.254 |
| | PC1 | 0.037 | 0.102 | 0.365 | 0.721 |
| $pCOC_{INT}$ | **PC2** | **0.434** | **0.120** | **3.948** | **0.001** |
| | PC3 | −0.205 | 0.175 | −1.169 | 0.260 |
| | PC1 | 0.245 | 0.172 | 1.424 | 0.185 |
| $pBOC_{INT}$ | PC2 | 0.120 | 0.205 | 0.588 | 0.570 |
| | PC3 | −0.618 | 0.362 | −1.709 | 0.118 |

**Notes.**
Bold indicates a significant value at the $p < 0.05$ level.

MSBL7), which belong to the dominant order Desulfobacterales, were predominant (Fig. S2). In contrast, family groups in Woeseiaceae (major genus *Woeseia*), Halieaceae (major genus *Halioglobus*), and Thiotrichaceae (which were not assigned to genus level) were predominant in the Gammaproteobacterial class (Fig. S3). Among the bacterial groups, Woeseiaceae was recently reported to be a cosmopolitan and abundant core member of the bacterial communities in marine sediments globally (*Dyksma et al., 2016*; *Mußmann et al., 2017*). These bacterial groups are frequently reported to be

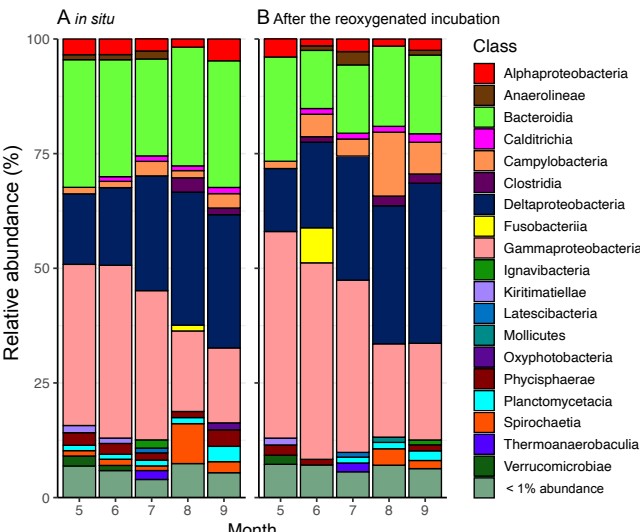

**Figure 4** **Stacked bar graphs of bacterial community composition from (A) the start and (B) the end of the reoxygenated incubation experiment for INT reduction measurement.** The partial 16S rRNA gene amplicons show the relative abundance (%) of counts. Groups demonstrating < 1% abundance were summarized as "< 1% abundance".

present in marine sediment surfaces (*Kawahara et al., 2009*; *Zinger et al., 2011*; *Dyksma et al., 2016*; *Jochum et al., 2017*). Therefore, the bacterial community composition in the surface sediment of Omura Bay demonstrated the ubiquity of the organisms (e.g., Desulfobacteraceae and Woeseiaceae) found in marine sediments.

According to our previous study (*Mori et al., 2018a*; *Mori et al., 2018b*), temporal shifts in the bacterial community in the surface sediment of Omura Bay were characterized by decreases in the relative abundance of the ASVs affiliated with Gammaproteobacteria toward less oxic conditions; conversely, those affiliated with Deltaproteobacteria exhibited the opposite results. The present results were essentially consistent with this trend, although there was a peak in the relative abundance of Deltaproteobacteria under the oxic condition in September 2012 (Fig.4). A strong typhoon, *Sanba*, had passed near Omura Bay on the day before we collected the sediment cores in September 2012. Due to the strong wind (>12 m/s on average) of the typhoon, water column of entire bay must have been mixed rigorously and oxygenated the bottom water. This mixing event is likely to be largely responsible for the contradiction between bottom water oxygen levels and the peak in Deltaproteobacteria. In contrast to a large fluctuation of DO in the water column under the typhoon, relative abundance of Deltaproteobacteria in the sediment seemed less affected.

Following the incubation (24 h) of sediment core samples, more than 90% of the ASVs (comprising 90.2%–96.6% of the total reads) did not substantially change their relative abundance (<0.1% of the abundance). Consistent with this notion, the ASVs were tightly clustered by sampling months (Fig. 5). However, it was found that several ASVs exhibited large increases or decreases during the incubation period. The majority of the increased ASVs belonged to the Arcobacteraceae, Fusobacteriaceae and Halieaceae family groups,

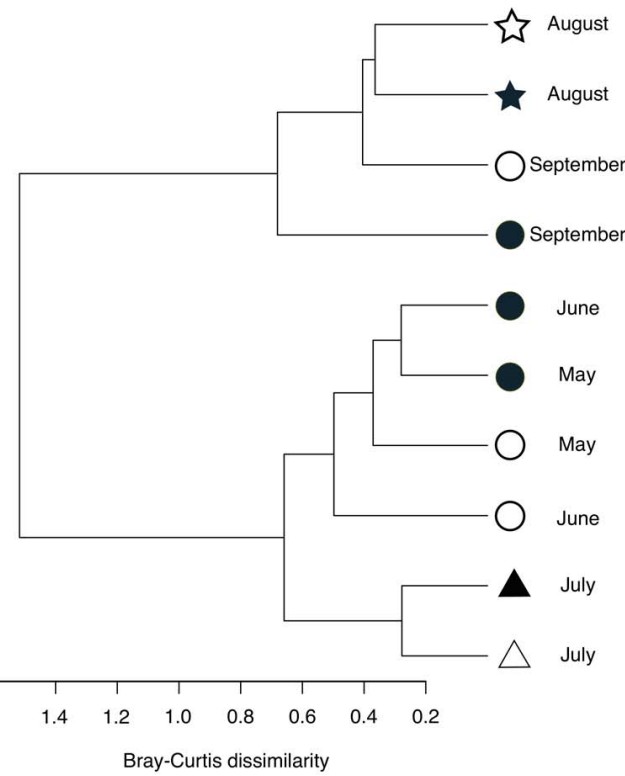

**Figure 5  Hierarchical clustering of the bacterial community composition in the sediment samples.**
The distance matrix was calculated based on the Bray–Curtis dissimilarity, and clusters were calculated using Ward's method. Open symbols indicate samples that were collected at the start of the incubation experiment. Closed symbols indicate samples that were collected at the end of incubation experiment. The symbols indicate the oxygen conditions in the bottom water (circle, oxic conditions; triangle, dysoxic conditions; and star, anoxic conditions).

while those of the decreased ASVs were affiliated with the Flavobacteriaceae, Saprospiraceae and Spirochaetaceae family groups (Fig. 6). The relative abundance of the Arcobacteraceae family group major genus *Arcobacter* increased from <1% to 12.9% within 24 h in the sediment cores that were retrieved under anoxic bottom water conditions (August 24, 2012) (Table S2). It was interesting to note that the increase in the relative abundance of *Arcobacter* ASVs coincided with the highest pBOC (Fig. 2). The *Arcobacter* comprise a group of physiologically diverse bacteria that exhibit anaerobic dissimilatory manganese reduction (*Vandieken et al., 2012*) or aerobic sulfide oxidation (*Wirsen et al., 2002*; *Sievert et al., 2007*). Given that these bacterial groups have been found in marine sediments under suboxic conditions and were reportedly engaged in the sulfur cycles (*Wirsen et al., 2002*; *Sievert et al., 2007*), the surge in pBOC$_{INT}$ in August 2012 could be ascribed to the aerobic oxidation of reduced sulfur compounds by *Arcobacter*. Consistent with this interpretation, a recent report on the metatranscriptomics of anoxic marine sediment revealed *Arcobacter* was highly responsive to oxygenation event and upregulated expression level of the genes encoding for respiratory proteins (*Broman et al., 2017*). Therefore, it is very likely that the anoxic sediment of Omura Bay harbored an *Arcobacter* population capable of rapidly

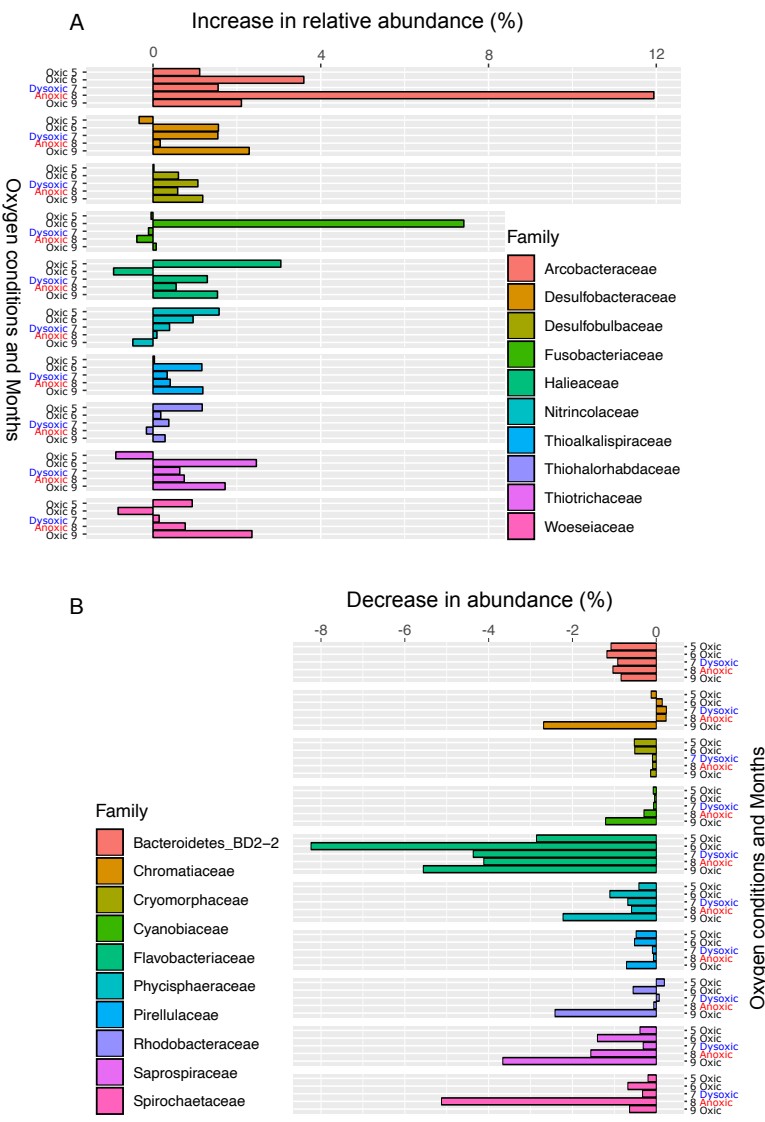

**Figure 6  Bar plot of the top ten families showing an increase (A) or decrease (B) 24 h after oxygenation in 2012.** The percentage of change was calculated by subtracting the relative abundance at the end of the incubation from that of *in situ* sediment samples. Dissolved oxygen condition at the site has been showing, according to the criteria used by *Wright, Konwar & Hallam (2012)*: Oxic (>90 µM O2), dysoxic (90–20 µM O2), suboxic (20–1 µM O2), and anoxic (<1 µM O2).

growing in response to the replenishment of DO. Besides *Arcobacter*, it is reasonable to assume the other bacterial groups whose relative abundance increased notably during the incubation period, such as Fusobacteriaceae and Halieaceae (Fig. 6), also contributed to the pBOC$_{INT}$.

On the other hand, it was difficult to associate the dominant bacterial groups with potential respiratory activities. Even though a majority of the predominant components of the bacterial community, such as the Woeseiaceae group, is likely to play a role in sediment metabolism, the present results provided no further insights to support this inference. One

possibility for the contribution of dominant groups to the SOC is that highly abundant anaerobic sulfate reducing bacteria might have affected the seasonal dynamics of pBOC via their aerobic respiration. The sulfate reducing bacteria inhabiting the sediment surface generally possess abilities to cope with high oxygen stress, and some of those are capable of aerobic respiration (*Cypionka, 2000*; *Brioukhanov, Pieulle & Dolla, 2010*; *Zhou et al., 2011*), which possibly contributes to pBOC$_{INT}$ during the oxygenation experiment. Indeed, the maximum pBOC$_{INT}$ observed in August 2012 was the timing when cells abundance was peaked and sulfate reducing Deltaproteobacteria, including the groups capable of aerobic respiration such as *Desulfobacter* (*Dannenberg et al., 1992*), was highly abundant in the surface sediment (Fig. 2B, Table 1, Fig. S2). It is not clear whether electrogenic sulfur oxidation by cable bacteria plays roles in Omura Bay, as we did not detect the cable bacteria genus (i.e., Candidatus Electrothrix). It will be worth investigating its presence using other methods, such as fluorescent *in situ* hybridization, in any future studies.

## CONCLUSION

We quantitively demonstrated a significant relationship between INT reduction and potential SOC in Omura Bay. Based on the relationship, it was revealed that not only pCOC$_{INT}$ but also pBOC$_{INT}$ increased during the hypoxic period of the bay, highlighting much greater contribution of pBOC to the pWOC in hypoxic sediment than previously thought. We further demonstrated *Arcobacter* could play vital roles in pBOC during hypoxia. This bacterial group may rapidly grow at the sediment surface in response to the episodic dissolved oxygen supply to otherwise hypoxic bottom. Sulfate reducing bacteria were dominant in the sediment surface under anoxic conditions and possibly contributed to the pBOC via aerobic respiration when dissolved oxygen was temporarily supplied. Clearly, more detailed studies are necessary to clarify the contribution of phylogenetically diverse sedimentary bacteria to the seasonal dynamics of coastal SOC.

### Funding
This work was supported by JSPS KAKENHI (No. 22580202 and 17H03854) to Minoru Wada, and JSPS KAKENHI (No. 19K23685) and the Kurita Water and Environment Foundation (No. 18B075) to Fumiaki Mori. The funders had no role in study design, data collection and analysis, decision to publish, or preparation of the manuscript.

### Grant Disclosures
The following grant information was disclosed by the authors:
JSPS KAKENHI: 22580202, 17H03854, 19K23685.
Kurita Water and Environment Foundation: 18B075.

### Competing Interests
The authors declare there are no competing interests.

## Author Contributions

- Fumiaki Mori conceived and designed the experiments, performed the experiments, analyzed the data, prepared figures and/or tables, authored or reviewed drafts of the paper, and approved the final draft.
- Yu Umezawa performed the experiments, authored or reviewed drafts of the paper, and approved the final draft.
- Ryuji Kondo performed the experiments, analyzed the data, authored or reviewed drafts of the paper, and approved the final draft.
- Gregory N. Nishihara analyzed the data, prepared figures and/or tables, authored or reviewed drafts of the paper, and approved the final draft.
- Minoru Wada conceived and designed the experiments, performed the experiments, authored or reviewed drafts of the paper, and approved the final draft.

## DNA Deposition

The following information was supplied regarding the deposition of DNA sequences:

DNA sequences are available in the DDBJ Sequence Read Archive: DRA008748 (DRX177081–DRX177090).

## Data Availability

The raw measurements are available in the Supplementary Tables.

## Supplemental Information

Supplemental information for this article can be found online at http://dx.doi.org/10.7717/peerj.11836#supplemental-information.

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
