# Peer review of "Potential oxygen consumption and community composition of sediment bacteria in a seasonally hypoxic enclosed bay"

_PeerJ, doi:10.7717/peerj.11836_

## Round 0.1 · original submission · Minor Revisions

I concur with both reviewers, that minor revisions are required. Please pay close attention to the detailed reviewers comments.

Reviewer 1 ·

Basic reporting

Yes, English meaning and grammar is clear. One general comment is that there are a lot of abbreviations and keeping track of them mentally is challenging and detracts from the readers capacity to interpret the research. If you are not restricted by word count I would suggest limiting their use by using whole words.

Literature of the use of INTR is out of the scope of my expertise.
Literature on sediments and microbial communities is relevant.

Article is structured into Introduction, methods and combined results/discussion

Figures and tables support primary results. Figure 5 labelling – hexagon would be more appropriately named star. Sequencing data is available
-
Bacterial names need to be italicized – see legend Fig 6 and throughout text

More specific comments:
The strength in this study was in the reporting of the chemical data. Associating changes in the composition of the microbial community to the biological consumption of oxygen and the chemical consumption could be improved. My interpretation is that the biological (BOC) is a measure of direct oxygen consumption and that chemical (COC) is a measure of indirect biological consumption mixed with other reductants present within the sediments. With this in mind I find Figure 2 interesting as it implies that the majority of oxygen consumed by the sediment is not directly via respiration (microbial or microphytobenthos). The primary process reported here (but not discussed at length) appears to be via indirect biological processes that increase the proportion of reduced compounds (pCOBINT), which then consume available oxygen. Although difficult to achieve I would have liked the authors to discuss how these data, especially changes in the composition of the microbial community, could be contributing to this. For example, does increased pBOCINT co-occur with a decreased relative abundances of known anaerobes (who could be producing the reduced products)? or in identifying and temporally tracking the strict aerobes (which might be a simpler way to achieve the same thing)?

Line 393-396. Do you have any taxonomic support within your dataset from 2012 for this statement “As the highest pBOCINT in 2012 and 2013 (2.82 µmol INTF g−1 day−1 in 2012 and 4.16 µmol INTF g−1 day−1 in 2013) coincided with the timing of the highest bacterial count in each year (1.7 × 1011 cells/g sediment in 2012 and 3.2 × 1010 cells/g sediment in 2013), it was likely that the increases in facultative anaerobes or aerobes contributed to the surge of the pBOCINT.” One way to analyze this would be to apply a time-series model. This could be achieve using a log-ratio transformation or a random forest approach?

Line 458 – I agree that making these linkages is important for the interpretation of this study and doing it more would increase the conclusions and relevance to the stated question. However, changes in relative abundances between treatments can not be compared directly without controlling for change in the relative abundances of taxa around them. Although with their own biases, tools exist to achieve this such as deseq and ancom. See: “It was interesting to note that the increase in the relative abundance of Arcobacter ASVs coincided with the highest pBOC (Fig. 2).”

Figure 6 – Instead of labelling by month sampled it would be interesting to label these data with the oxygen availability in situ. As the impact of oxygenation after 24h will change based on the pre-existing oxygen conditions of the sediment.


Line 62 – define NH4+, Fe2+, and HS-,
Line 127-129 – is this required? (This research was conducted by Fumiaki Mori in partial fulfillment of the requirements for a Ph.D. from Nagasaki University (Mori, 2018))
Line 385 – which part of fig 2 are you referring to here?
Line 396 – add table 1 reference to bacterial cell counts

Experimental design

no comment

Validity of the findings

no comment

·

Basic reporting

The manuscript is clear and professional. Some minor observations:

Line 29, including the frequency of the monitoring.

Line 60: grammatical error, change "in" to "of"

Line 151: delete “of”

Line 357: “BIR” is not defined in the whole document.

Experimental design

In the equation in line 183, it is unclear if the incubation time changed between the experiments.

Line 190, ¿What kind of bottles? ¿ODB, PC?

The GAM that was adjusted is not shown. I suggest that the authors show the GAM equation just as they did with the GLM (Eq. 2).

Line 238: What is the proportion of isopropanol and methanol?

Line 240: Delete: “(See Comparison between….)”

Validity of the findings

Line 326-327: Include the reference for continuous cultures of natural communities of marine prokaryotes: Villegas-Mendoza et al. 2019. "The Chemical Transformation of the Cellular Toxin INT (2- (4-Iodophenyl) -3- (4-Nitrophenyl) -5- (Phenyl) Tetrazolium Chloride) as an Indicator of Prior Respiratory Activity in Aquatic Bacteria" ·.

The authors reasonably argue in lines 342-350 how the incubation of sediments with INT circumvent the limitations (toxicity, INT recovery) reported for the quantification of INTF in planktonic samples.

Line 372: Why did the effect of formalin only occur in 5 of the 20 samples?

Additional comments

The manuscript includes relevant information that contributes to the quantification of respiratory processes in sediments. The relationship of temperature and preservation of organic matter during anoxic conditions seems very interesting (Fig. 3). Although it is mentioned that the temperature is a determining factor of the oxygen consumption of the sediments. It would be enriching for discussion by including a more detailed explanation of the effect of temperature and substrate concentration as a driver of metabolic parameters in the identified microbial community.

---

## Round 0.2 · accepted · Accept

I agree with both reviewers that this manuscript is now ready for publication. Thank you for your careful attention to the reviewers' comments.

Reviewer 1 ·

Basic reporting

No changes

Experimental design

No changes

Validity of the findings

No changes

Additional comments

No changes

·

Basic reporting

I appreciate the author’s effort in providing answers to my assessment.

Experimental design

The experimental design is robust and detailed. The author has considered the previous comments. The methods provide specific information for their replication.

Validity of the findings

The authors added the information requested in the previous review. The discussion and conclusions are clear and concise and correspond to the information provided in the results.

Additional comments

I appreciate the author’s effort in providing answers to all my questions.